# Intelligent Fault Diagnosis Method Using Acoustic Emission Signals for Bearings under Complex Working Conditions

**Minh Tuan Pham** [1], **Jong-Myon Kim** [2] **and Cheol Hong Kim** [3,*]

1   School of Electronics and Computer Engineering, Chonnam National University, Gwangju 61186, Korea; 196325@jnu.ac.kr
2   School of IT Convergence, University of Ulsan, Ulsan 44610, Korea; jmkim07@ulsan.ac.kr
3   School of Computer Science and Engineering, Soongsil University, Seoul 06978, Korea
*   Correspondence: cheolhong@ssu.ac.kr; Tel.: +82-2-820-0674

**Abstract:** Recent convolutional neural network (CNN) models in image processing can be used as feature-extraction methods to achieve high accuracy as well as automatic processing in bearing fault diagnosis. The combination of deep learning methods with appropriate signal representation techniques has proven its efficiency compared with traditional algorithms. Vital electrical machines require a strict monitoring system, and the accuracy of these machines' monitoring systems takes precedence over any other factors. In this paper, we propose a new method for diagnosing bearing faults under variable shaft speeds using acoustic emission (AE) signals. Our proposed method predicts not only bearing fault types but also the degradation level of bearings. In the proposed technique, AE signals acquired from bearings are represented by spectrograms to obtain as much information as possible in the time–frequency domain. Feature extraction and classification processes are performed by deep learning using EfficientNet and a stochastic line-search optimizer. According to our various experiments, the proposed method can provide high accuracy and robustness under noisy environments compared with existing AE-based bearing fault diagnosis methods.

**Keywords:** fault diagnosis; bearing fault; machine health monitoring; acoustic emission signals; convolutional neural network

---

## 1. Introduction

High-power, heavy, and large-size industrial electric motors play a vital role in the production process at factories. Therefore, these electric motors should be strictly monitored using advanced equipment. When the fault status of these machines is diagnosed accurately, finding specific components causing the fault quickly can help factories reduce downtime and repair costs and maintain the quality of products. In common fault-diagnosis (FD) processes, the signals which contain the machine health status are acquired and transferred to the server system in which they are analyzed to diagnose faults. Although this approach is expensive compared with the use of handheld FD instruments using the diagnosis method for bearing faults [1], it can improve reliability, which is undeniable in machine monitoring, especially for essential machines.

Bearing faults account for approximately 50% of faults related to electrical motors [2–4]. Therefore, many studies have focused on bearing FD using various types of signals acquired from sensors mounted on electric motors, especially vibration signals. In the last decade, with the advent of machine learning techniques, bearing FD methods using vibration signals have achieved high accuracy under specific conditions [5–7]. However, in realistic scenarios such as diagnosis of very early developed damage [3], diagnosis of very slow rotating machinery, and monitoring under high-vibration environments, acoustic

emission (AE) signals can be more useful than vibration signals. Therefore, AE-based FD methods can be mainly considered in fault monitoring systems.

Traditional digital signal processing-based FD methods such as envelope analysis achieve high accuracy only when the bearing speed is constant or changes slightly. Otherwise, when diagnosing bearing faults under variable rotational speed (e.g., in a wind-turbine system, the bearing speed continuously changes during operation [8]), acquired AE signals tend to be non-stationary. With the potential advantage of showing the change of signal frequency components with time, the time–frequency $(t, f)$ analysis showed its efficiency. Time–frequency $(t, f)$ representation can reveal more features of temporal localization of signal spectral components, resulting in better efficiency than frequency domain analysis and time-domain analysis. Zang et al. proposed a method using continuous wavelet packet transform (CWT) to extract compressive waves that contain the consistent speed of a given frequency band to locate faulty planet gear in a wind turbine gearbox [9]. Moreover, the development of convolutional neural networks (CNN) has provided an automatic and incredibly efficient tool for extracting and selecting useful features containing fault information from time–frequency AE signal representations. Recent methods for compound bearing fault diagnosis under variable speeds used generic CNN architecture with three convolution layers based on Lenet-5 and enhanced the CNN training process by a stochastic diagonal Levenberg–Marquardt algorithm [10,11]. In these methods, AE signals are converted into two-dimensional spectral energy distribution maps to feed the CNN. Those methods achieved high accuracy, but in noisy environments, this simple CNN architecture is hard to extract faults information effectively.

With the higher diagnosis accuracy of CNN-based methods compared to traditional signal processing methods, applying a more suitable CNN model in bearing FD is essential to analyze the information from signals and then, enhance accuracy, consistency, and computation efficiency. In this paper, we propose a new method to achieve high FD accuracy in diagnosing multiple faults under various rotational speeds and noisy environments. In addition, the degradation level of each type of fault can be predicted in our proposed method. Our method employs a state-of-the-art CNN model, EfficientNet [12], to automatically extract features from the spectrogram representation calculated from acquired AE signals in the time–frequency domain. For applying EfficientNet to a fault classification process effectively, the optimizer named stochastic line search (SLS) [13] is also adopted in the training stage, resulting in improved convergence ability, decreased training time, and reduced number of training samples.

## 2. Proposed Bearing Fault Diagnosis Method using Acoustic Emission Signals

In this section, a novel method for diagnosing bearing faults under variable rotational speeds using AE signals adopting Short-time Fourier Transform (STFT), CNN model, and SLS is presented. Figure 1 shows the overall process of the proposed method. The proposed method consists of three major stages. In the beginning, the acquired raw AE signals are split into 0.1 s fixed cycle signal segments. Then, STFT is applied to AE signal segments to create spectrogram images as the representation method in the time–frequency domain. After that, image normalization is applied to the created spectrogram images as a way of enhancing their representation ability. Besides, the spectrogram images are standardized to fit the CNN model. Secondly, the EfficientNet CNN model randomly initializes its parameters and is pre-trained by the spectrogram images. During the training process, the SLS optimizer is used to increase the probability of convergence and accuracy in fault classification. Then, the trained CNN model is used to classify bearing faults automatically in the final stage.

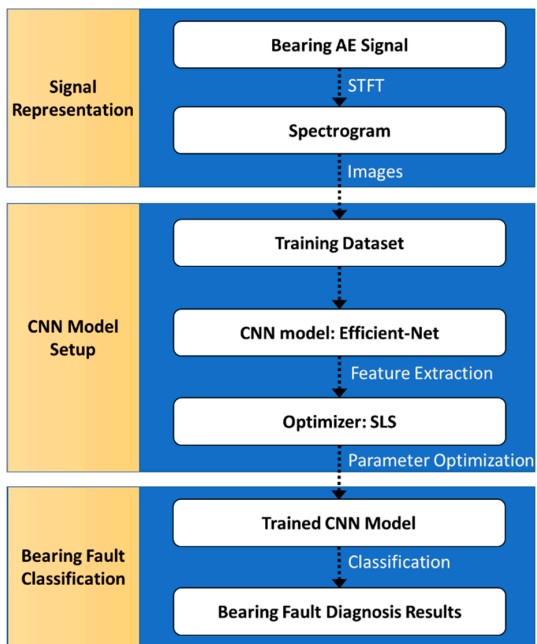

**Figure 1.** Proposed bearing fault diagnosis method using acoustic emission (AE) signals.

### 2.1. Short-Time Fourier Transform (STFT)

For diagnosing bearing faults in variable rotational speeds, most fault frequencies are not determined because these frequency bands directly depend on rotational speeds, where STFT can provide the information to analyze. STFT provides time–frequency analysis which can solve two notable problems that time or frequency domain analysis face: (1) single domain analysis has difficulty in illustrating time-dependency of the spectrum for nonstationary signals; (2) Fourier Transform (FT) is not compatible with representing transient signals due to its non-periodic characteristics. For the appropriate analysis and synthesis of nonstationary signals, Short-Time Fourier Transform (STFT) is based on a series of overlapped Fast Fourier Transforms (FFTs) applied for signal segmentation [14].

In transient states such as acceleration periods, start-up, and shutdown, the acquired AE signals can contain more suitable information about bearing conditions compared to steady stages of machines. For this reason, the STFT, which can represent the transient information from AE signals, is adopted in our proposed method. The spectrogram is used as a visual representation method of the AE signals in both time and frequency domains, using a color scale of the image to indicate the frequency's amplitude.

### 2.2. Creating and Processing Bearing Faults Spectrograms

Optimizing STFT depends on (1) finding compatible window size; (2) overlap between segments (hop size), which affects the density in time; (3) zero-padding for FFT calculation; and (4) choosing a suitable signal segment size. Those STFT parameters can be determined by the trade-off relation between frequency resolution and time resolution. For applying the STFT to bearing fault signals efficiently, the STFT matrix is determined by a new routine with the MATLAB tool to achieve high accuracy and computational efficiency [15]. The segment duration of the signal is also considered based on the fault frequency range, which depends on the rotational speed range. Besides, the parameters are chosen to ensure that signal statistic features are as steady as possible. The window length (*wlen*) is set to 1024, and the hop size is set to *wlen*/4, as referred in various experiments [15]. Figure 2 shows our AE signal spectrograms at various rotational speeds (250, 300, 350, 400, 450, and 500 RPM) for three kinds of single faults (inner race faults, outer race faults, and roller faults). Similar results are obtained for compound faults (inner and outer raceway cracks (BCIO), outer and roller cracks (BCOR), inner and roller cracks (BCIR), inner–outer–roller cracks (BCIOR)) and normal condition (BNC).

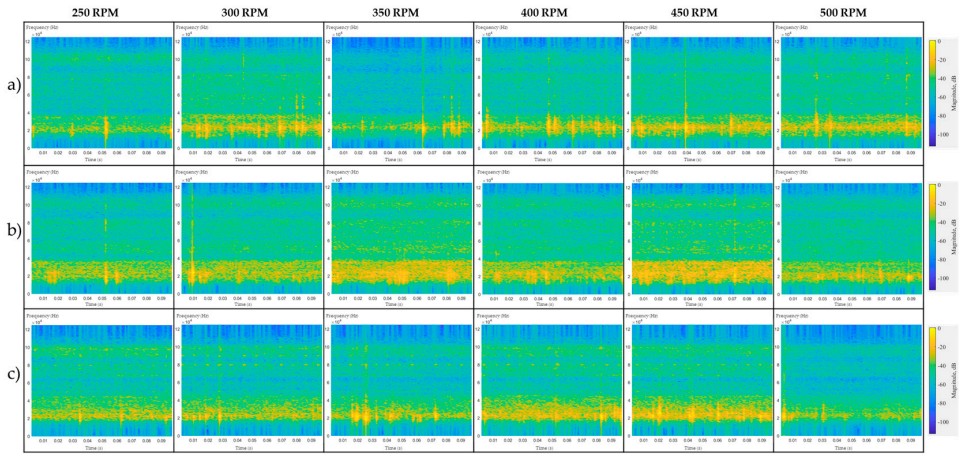

**Figure 2.** Spectrograms of bearing AE signals at six different shaft speeds: (**a**) inner race bearing faults; (**b**) outer race bearing faults; and (**c**) roller faults.

Spectrogram image normalization:

Normalization [16] is applied to change the range of pixel intensity values of created spectrogram images to make those images more suitable to analyze. Figure 3 shows the difference between spectrogram images before and after the normalization. Through the normalization process, a tensor image is calculated by the mean and standard deviation; the output of each image's channel is illustrated in Equation (1):

$$Output[channel] = \frac{Input[channel] - mean[channel]}{std[channel]} \tag{1}$$

where $mean[channel]$ is the sequence of the mean for channels and $std[channel]$ is the sequence of standard deviations for channels. After the normalization, spectrogram images are resized to $224 \times 224$ to fit the input layer size of the used CNN model. The image normalization is used as image pre-processing to yield better input data. Therefore, training time and the number of samples used for training the CNN model can be decreased through image normalization.

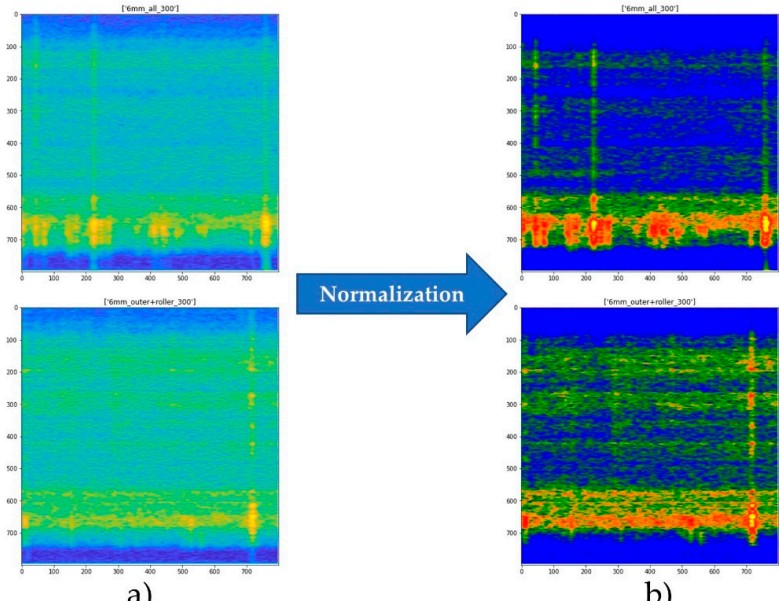

**Figure 3.** Bearing fault spectrogram images: (**a**) before normalization; (**b**) after normalization.

### 2.3. EfficientNet CNN Model for Bearing Fault Diagnosis

Scaling up the CNN from a baseline network is broadly implemented to achieve superior precision [12]. EfficientNet is based on this principle to create a CNN network to achieve better accuracy and efficiency [12]. A pre-existing CNN structure is defined by the function $N = \underset{i=1...s}{\odot} F_i^{L_i}(X_{\langle H_i, W_i, C_i \rangle})$, where $F_i^{L_i}$ denotes layer $F_i$, which is repeated $L_i$ times in stage $i$, and $\langle H_i, W_i, C_i \rangle$ represents the shape of input tensor $X$ in layer $i$ [12]. Functional changes are made in the scaling dimension parameters, including depth $d$ (i.e., the number of CNN layers in the network), width $w$ (size of the CNN layers), and resolution (resolution of the input image). EfficientNet provides a constraint in selecting these three parameters, called the compound scaling method, which uses compound coefficient $\theta$ to uniformly scale the network width, depth, and resolution in a reasonable manner. Equation (2) illustrates those parameters and their constraints.

$$
\begin{aligned}
depth &: d = \alpha^\theta \\
width &: w = \beta^\theta \\
resolution &: r = \gamma^\theta \\
&\alpha \cdot \beta^2 \cdot \gamma^2 \approx 2 \\
&\alpha \geq 1, \beta \geq 1, \gamma \geq 1
\end{aligned}
\tag{2}
$$

In Equation (2), $\alpha$, $\beta$, and $\gamma$ are constants that can be found using a small grid search. Intuitively, $\theta$ is a user-specified coefficient that controls how many resources are available for model scaling, whereas $\alpha$, $\beta$, and $\gamma$ specify how to assign these extra resources to the network width, depth, and resolution, respectively.

Figure 4 shows the architecture of EfficientNet-B0 developed based on mobile inverted bottleneck MB-Conv [17], where it is also added the squeeze-and-excitation optimization [18]. MB-Conv is designed to run deep networks on personal mobile devices. Therefore, MB-Conv is not only effective but also requires low energy consumption. The EfficientNet architecture is designed using depth-wise separable convolution as efficient building blocks, linear bottlenecks between layers, and shortcut connections between the bottlenecks. EfficientNet is a group consisting of 8 models, B0–B7.

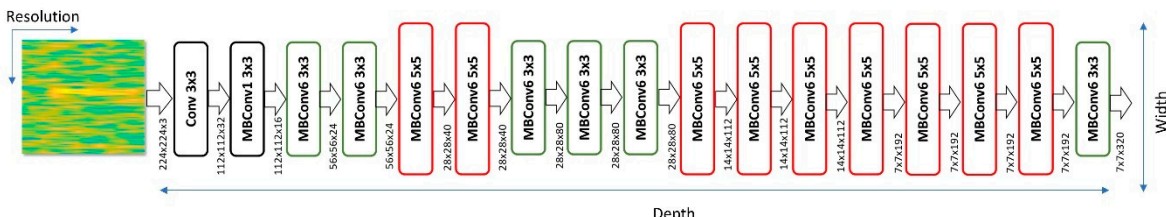

**Figure 4.** Scalability of EfficientNet.

The EfficientNet-B0 model was used by selecting the parameters based on the following process. First, we let $\theta = 1$ and used a grid search for $\alpha$, $\beta$ to rescale the model from B0 to B1. After that, we had the set of parameters $\alpha = 1.2$, $\beta = 1.1$, and $\gamma = 1.15$. Then, we fixed $\alpha$, $\beta$, and $\gamma$ as constants and scaled up the baseline network under different $\theta$ values using Equation (2) to obtain EfficientNet-B1 to B7. In this study, EfficientNet-B0 was used because it can sufficiently extract useful bearing fault information and does not use redundant computational resources.

### 2.4. Stochastic Line Search Optimizer

Gradient Descent (GD) is one of the most well-known and standard algorithms to optimize CNN models [19]. There are many GD-based variants, such as Stochastic Gradient Descent (SGD), Adam, and Momentum [19]. However, these optimizers are used as black boxes. In general, different optimizers need to be applied to find out the most compatible one based on their strengths and weaknesses. The Adamax optimizer was utilized to strengthen the training process after trying various

optimizers (SGD, Adam, and Adamax) [20]. Therefore, we can know that the convergence ability of CNN models can be enhanced when a compatible optimizer is adopted.

Stochastic gradient descent (SGD) and its variants are highly effective in modern machine learning models [21]. In any case, the gradient methods are antagonistically influenced by two challenges: Firstly, their execution depends significantly on the setting of learning rate. Secondly, frequently updating the steps taken towards the minima shows higher probability to lead the gradient descent into other directions compared to across all training samples (due to noisy steps, it can take longer to converge to the minima of loss function).

A stochastic line search was developed based on traditional line-search methods to find a suitable step size (learning rate) without pre-setting for SGD while training over-parametrized models. To do that, the line search approach seeks a downward trend directory of the objective function, then infers the step size, which corresponds to the length to move along that direction.

SGD computes the gradient of the loss function randomly fed by one sample or by a minibatch of chosen training samples $i_k$ in iteration $k$. It then performs an update step as $w_{k+1} = w_k - \eta_k \nabla f_{ik}(w_k)$ corresponding to loss function descent, where $w_{k+1}$ and $w_k$ are the SGD iterates, $\eta_k$ is the step size, and $\nabla f_{ik}(\cdot)$ is the average loss function gradients computed at iteration $k$. Each stochastic gradient $\nabla f_{ik}(w)$ is assumed to be equivalent (e.g., $E_i[\nabla f_i(w)] = \nabla f(w)$ for all $w$).

The Armijo line search is a criterion for deterministically setting the step size of gradient descent [13]. At iteration k, the Armijo line search performs computations to choose a step size that satisfies the following condition, where c > 0 is a hyperparameter:

$$f_{ik}(w_k - n_k \nabla f_{ik}(w_k)) \leq f_{ik}(w_k) - c \cdot \eta_k \left\| \nabla f_{ik}(w_k) \right\|^2 \tag{3}$$

The implementation of the SLS process in our proposed bearing fault diagnosis method can be summarized as follows.

1.　Compute the gradients $\nabla f_{ik}(w_k)$ for a given training batch.
2.　Search for a step size $\eta_k$ that satisfies the stochastic Armijo line search condition.
3.　Use the step size and update the model parameters with SGD:

$$w_{k+1} = w_k - \eta_k \nabla f_{ik}(w_k) \tag{4}$$

## 3. Experimental Implementation

The AE signals from the bearings are obtained from the experimental testbed [3], as shown in Figure 5.

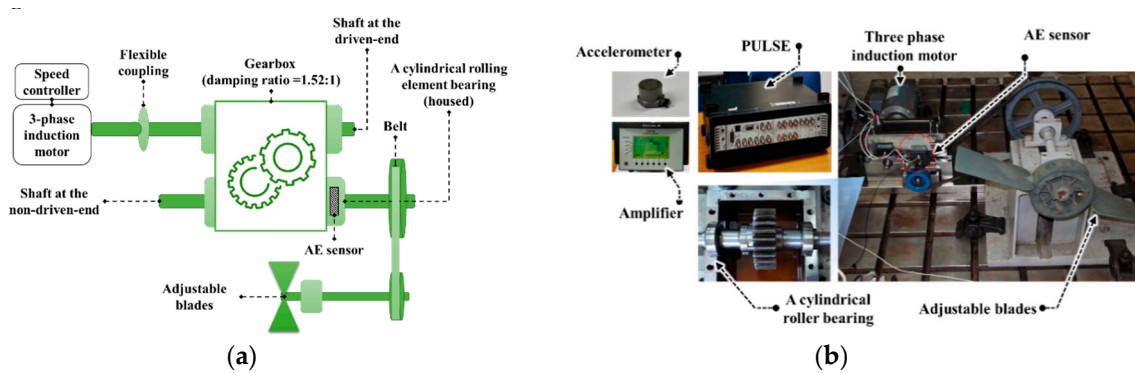

**(a)**　　　　　　　　　　　　　　　　　　　　**(b)**

**Figure 5.** Testbed for collecting AE signals from bearings with compound faults: (**a**) Testbed diagram; (**b**) Real testbed components positions.

The system contains two shafts: drive-end and non-drive-end shafts. These shafts are connected by a gearbox with a ratio of 1:1.51. In addition, rolling-element bearings are used to fasten both ends

(FAG NJ206-E-TVP2 Motion Industries, Birmingham, AL). The system's main power comes from a three-phase induction motor that operates at different speeds: 250, 300, 350, 400, 450, and 500 RPM. The AE data-acquisition facilities are attached to the bearings in the non-drive-end shaft. Seven bearing faults are determined with three different crack lengths (3, 6, and 12 mm), and the cracks are created by using a diamond bit cutter at seven different locations: outer raceway (BCO); inner raceway (BCI); roller (BCR); inner and outer raceways (BCIO); outer raceway and roller (BCOR); inner raceway and roller (BCIR); and inner raceway, outer raceway, and roller (BCIOR), as depicted in Figure 6.

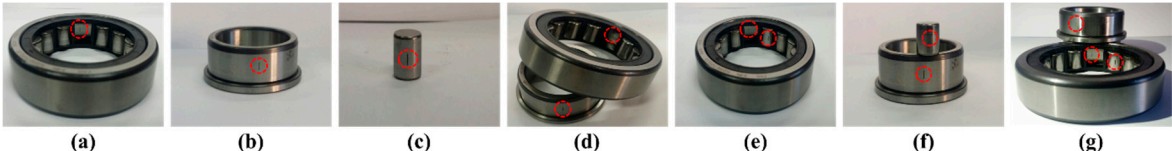

**Figure 6.** Single and compound bearing faults: (**a**) outer raceway (BCO); (**b**) inner raceway (BCI); (**c**) roller (BCR); (**d**) inner and outer raceways (BCIO); (**e**) outer raceway and roller (BCOR); (**f**) inner raceway and roller (BCIR); (**g**) inner raceway, outer raceway, and roller (BCIOR).

The data-acquisition system consists of a wideband AE sensor and PCI-2-based system, as described in Table 1. The system can capture the sampled AE signals at 250 kHz. The AE signals, which come from one normal condition (BNC) and seven defective bearings, are used to create three datasets. The first and second datasets are divided according to the crack length (3 and 12 mm), and the third dataset consists of all AE signals with three different crack lengths (3, 6, and 12 mm). In detail, each of these datasets is divided into three types of subsets: training subset, validation subset used for the training process, and testing subset used for evaluating our model performance. In the training process, besides the training subset, the validation subset is used during training to observe the model's learning ability. That helps avoid unexpected phenomena such as overfitting. The test subset and validation subset have the characteristics which are unseen by the model and are not used to update model parameters while training.

**Table 1.** Summary of AE signals acquisition system.

| | |
|---|---|
| AE sensor (PAC WSα) | Peak sensitivity (V/μbar): −62 dB<br>Operating frequency range: 100–900 kHz<br>Resonant frequency: 650 kHz<br>Directionality: ±1.5 dB |
| 2-channel PCI board | 18-bit 40 MHz A/D conversion<br>10M samples/s rate as only one channel is used (5M samples/s as two channels are used simultaneously)<br>Dynamic range: >85 dB<br>Sensor testing: AST build-in |

The samples of the training subset are independent of the rotational speed to the validation and testing subsets, as shown in Table 2. The training subset consists of spectrum images created from the AE signals collected at shaft speeds of 300, 400, and 500 rpm, whereas the validation and testing subsets consist of the AE signals acquired at shaft speeds of 250, 350, and 450 RPM. In each of Dataset 1 and Dataset 2, the total number of samples used for the training process (training subset and validation subset) is $N_{Classes} \times N_{Samples} \times N_{Speed}$ = 1440 samples, where $N_{Speed}$ = 6, $N_{Classes}$ = 8, and $N_{Samples}$ = 30 ($N_{Samples}$ is the total number of signal samples acquired at a specific bearing condition and specific shaft speed). Further, Dataset 3 consists of bearing fault samples in three degradation levels corresponding to three different crack sizes. Therefore, the total number of samples used for the training process is $N_{Classes} \times N_{Samples} \times N_{Speed}$ = 3960 AE signals, where $N_{Speed}$ = 6, $N_{Classes}$ = 22 (three crack levels for each of the seven bearing fault types and one normal condition), and $N_{Samples}$ = 30. In the testing process, 1200 samples for each class are used to evaluate the model performance.

**Table 2.** Evaluated dataset for bearing faults.

| Single and Compound Bearing Failures | | Rotational Speed (RPM) | Crack Size | | |
|---|---|---|---|---|---|
| | | | Length (mm) | Width (mm) | Depth (mm) |
| Dataset 1 | Training subset | 300, 400, 500 | 3 | 0.60 | 0.30 |
| | Testing subset (and Validation) | 250, 350, 450 | | | |
| Dataset 2 | Training subset | 300, 400, 500 | 12 | 0.60 | 0.50 |
| | Testing subset (and Validation) | 250, 350, 450 | | | |
| Dataset 3 | Training subset | 300, 400, 500 | 3 | 0.60 | 0.30 |
| | | | 6 | 0.60 | 0.50 |
| | Testing subset (and Validation) | 250, 350, 450 | | | |
| | | | 12 | 0.60 | 0.50 |

## 4. Experimental Results

### 4.1. Diagnosis Accuracy for Compound Bearing Faults

All of the experiments on the datasets were repeated ten times. These first experiments were conducted on Dataset 1 and Dataset 2. Table 3 shows the accuracy of the proposed FD method, where the results are compared to four existing AE-based FD methods for compound faults detection under variations in the rotational speed.

**Table 3.** Diagnosis accuracy of the proposed method for compound bearing faults compared to the existing methods.

| | Methodologies | Average Accuracy for Each Fault Type (%) | | | | | | | | ACA (%) |
|---|---|---|---|---|---|---|---|---|---|---|
| | | BCI | BCO | BCR | BCIO | BCIR | BCOR | BCIOR | BNC | |
| Dataset 1 | Proposed method | 96.00 | 94.12 | 100.00 | 100.00 | 100.00 | 100.00 | 100.00 | 100.00 | 98.77 |
| | [11] | 100.00 | 98.88 | 98.51 | 97.77 | 95.18 | 100.00 | 100.00 | 99.62 | 98.74 |
| | [10] | 66.60 | 100.00 | 100.00 | 100.00 | 89.10 | 99.20 | 99.20 | 99.60 | 94.20 |
| | [22] | 11.11 | 13.33 | 100.00 | 100.00 | 97.77 | 97.77 | 66.21 | 22.41 | 63.57 |
| | [3] | 19.62 | 47.40 | 75.18 | 47.03 | 59.62 | 30.74 | 49.76 | 58.62 | 48.49 |
| Dataset 2 | Proposed method | 100.00 | 100.00 | 100.00 | 100.00 | 100.00 | 100.00 | 100.00 | 100.00 | 100.00 |
| | [11] | 100.00 | 100.00 | 98.51 | 95.47 | 99.18 | 100.00 | 98.88 | 99.62 | 98.95 |
| | [10] | 100.00 | 100.00 | 91.80 | 98.10 | 99.20 | 99.20 | 100.00 | 99.20 | 98.40 |
| | [22] | 24.34 | 26.47 | 97.77 | 97.77 | 100.00 | 100.00 | 68.24 | 28.16 | 67.84 |
| | [3] | 7.03 | 70.00 | 66.66 | 79.62 | 5.92 | 44.81 | 74.07 | 62.96 | 51.38 |

Firstly, we compared two CNN-based FD methods [10,11] using the spectra of AE signals to create two-dimensional (2D) energy distribution maps (EDMs). The created EDMs fed a generic CNN based on Lenet-5 architecture to extract the bearing fault features. After extracting features, in [11], a hybrid ensemble MLP–SVM classifier is used to classify the faults from extracted features. On the other hand, in [10], classification is performed by multilayer perceptron classifiers. Additionally, a stochastic diagonal Levenberg–Marquardt algorithm is used while training to enhance the training process. In the existing FD method proposed in [3], Kang et al. defined fault features by various statistical parameters and complex envelope analysis in the time and frequency domain based on specific fault frequency bands calculated by a specific rotational speed. Then, an outlier-insensitive hybrid feature selection (OIHFS) is applied to determine which features are useful to feed the K-NN classifier. An approach

that constructs features using the characteristic frequencies of bearing faults was also proposed by Tra et al. in [22].

The compared FD accuracy according to applied bearing FD methods, as listed in Table 3, illustrates that the methods [3,22] that use manual feature calculation based on fault characteristic frequencies fail to adapt well under variable shaft speed. The average accuracies of the method in [22] and method in [3] are 63.57% and 48.49% for Dataset 1, respectively. Even in the case of big crack size (Dataset 2), the two methods achieved similar results. In contrast, the automatic methods that use CNN models to extract the representative feature image that contains bearing fault information under various defect frequencies provide better accuracy. One of those CNN-based existing FD methods [11] provides an average accuracy of up to 98.74% and 98.95% for Dataset 1 and Dataset 2, respectively. Therefore, we can know that the CNN-based FD methods can achieve high accuracy even when the rotational speed varies. Among these compared CNN-based methods, our proposed method achieves the best FD accuracy by adopting a compatible image representation method, a capable and sufficiently deep CNN model, and an SLS optimizer to avoid missing fault features. Especially, when the crack size is 12mm, the proposed method achieves 100% FD accuracy. The results of the fault classification task can be depicted by using confusion matrices, as shown in Figure 7.

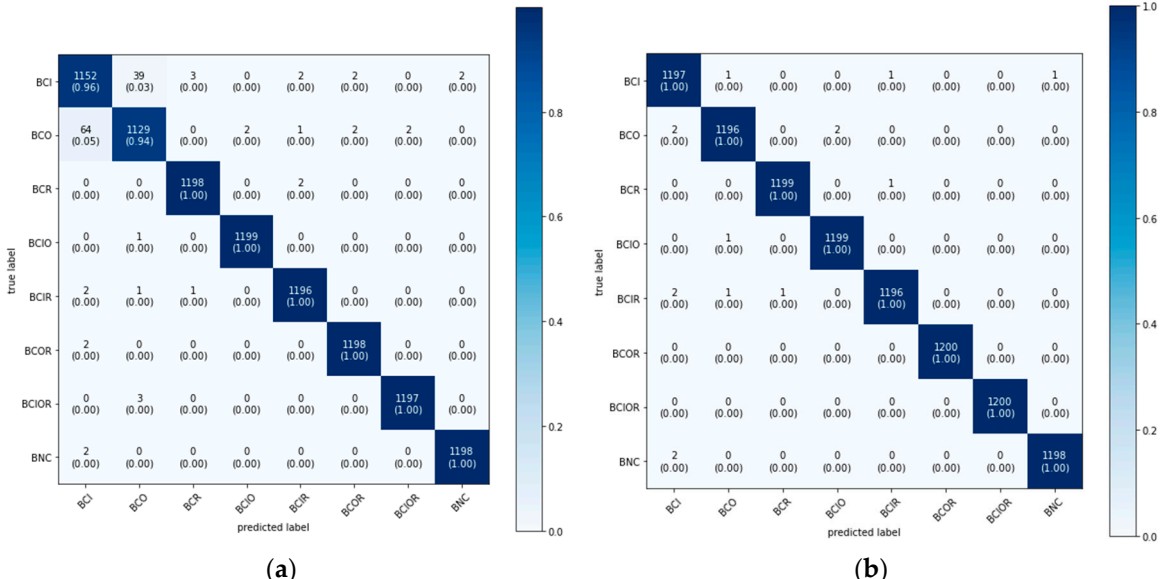

(**a**)          (**b**)

**Figure 7.** Confusion matrices for testing results; (**a**) Confusion matrix for Dataset 1; (**b**) Confusion matrix for Dataset 2.

Moreover, our bearing fault diagnosis method also can provide fast convergence ability (4 to 5 epochs for training). We analyzed the effectiveness of the adopted SLS optimizer by comparing it with the Adamax optimizer [23] in terms of convergence rate and convergence ability. The loss curve shows how well the model is learned from the input data. A higher value trend of the loss curve illustrates that the model's ability is worse at learning from training data. In Figure 8, we observe that the Adamax optimizer has good convergence results on both the training subset and the validation subset, and there is no overfitting phenomenon. That is because the Adamax optimizer is a variant of the Adam algorithm, which uses a momentum and infinity norm suitable for large amounts of data [23]. However, the SLS optimizer provides much faster convergence (in 4 to 5 epochs) and has better accuracy because it eliminates the disadvantages of choosing a suitable learning rate and choosing other Adam-based optimizers' parameters.

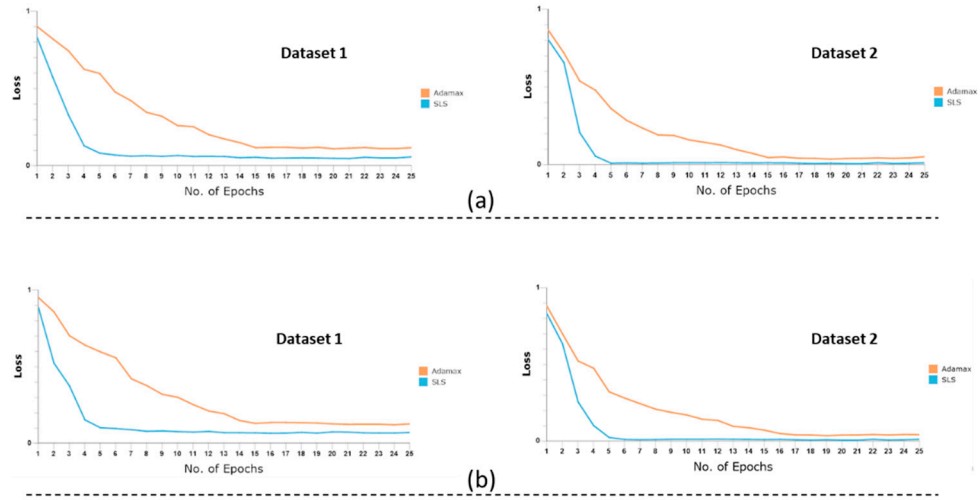

**Figure 8.** Comparison of two optimizers for two datasets: Adamax and SLS optimizer. (**a**) Training loss curves; (**b**) Validation loss curves.

### 4.2. Compound Fault Diagnosis in Noisy Conditions

The proposed method is also evaluated under noisy conditions. The AE signals (Dataset 1 and Dataset 2) are added with three white-noise levels (signal-to-noise ratio (SNR) = 0, 5, and 10 dB), where SNR is defined as the ratio of the power of a signal (meaningful input) to that of the background noise.

$$\text{SNR}_{dB} = 10\log_{10}\left(\frac{P_{signal}}{P_{noise}}\right) \tag{5}$$

The experimental results in noisy conditions are shown in Table 4, which indicate that in lower SNR values, the accuracy of the proposed method becomes relatively worse, but still provides high accuracy. When the SNR values are positive, the average accuracy is more than 89.85% and 92.11% for Dataset 1 and Dataset 2, respectively.

**Table 4.** Accuracy of proposed bearing fault diagnosis in noisy conditions.

| Dataset | SNR (dB) | Class Wise Accuracy (%) | | | | | | | | Average Accuracy |
|---------|----------|--------|--------|--------|--------|--------|--------|--------|--------|---------|
| | | BCI | BCO | BCR | BCIO | BCIR | BCOR | BCIOR | BNC | |
| | 10 | 100.00 | 100.00 | 100.00 | 100.00 | 92.86 | 100.00 | 100.00 | 87.50 | 97.55 |
| Dataset 1 | 5 | 85.71 | 92.31 | 100.00 | 100.00 | 100.00 | 100.00 | 100.00 | 91.67 | 96.21 |
| | 0 | 83.33 | 77.78 | 100.00 | 100.00 | 100.00 | 72.73 | 91.67 | 93.33 | 89.85 |
| | 10 | 100.00 | 100.00 | 100.00 | 91.82 | 94.12 | 100.00 | 100.00 | 100.00 | 98.24 |
| Dataset 2 | 5 | 80.00 | 100.00 | 100.00 | 80.00 | 100.00 | 92.31 | 100.00 | 100.00 | 94.03 |
| | 0 | 90.91 | 88.24 | 81.82 | 94.12 | 90.91 | 100 | 100.00 | 90.91 | 92.11 |

### 4.3. Classifying Compound Faults and Fault Degradation Levels

Further, we conducted experiments to evaluate the ability of our proposed method under complex conditions. This was diagnosing bearing faults and their degradation level corresponding to three different crack sizes (3, 6, and 12 mm) (e.g., inner and outer raceway fault (BCIO)—3 mm) under variable rotational speed. Dataset 3 was used in this experiment.

The results in Table 5 show that under noiseless conditions, the proposed method achieves an average accuracy of 98.21% for complex working environments. Under the presence of noise with positive SNR values, a few samples of the bearing conditions are misclassified, which deteriorates the overall prediction accuracy of the proposed model. However, the achieved accuracy is declined slightly to 96.08% when SNR is 10 dB and to 95.36% when SNR value is 5 dB. The worst accuracy is

93.33% when SNR value is 0, showing that the proposed model provides high diagnostic accuracy and stability under complex working environments.

**Table 5.** Average accuracy of the proposed method in diagnosing compound faults and different degradation levels.

|  | SNR (dB) | Average Accuracy (%) |
|---|---|---|
|  | No noise | 98.21 |
| **Dataset 3** | 10 | 96.08 |
|  | 5 | 95.36 |
|  | 0 | 93.33 |

Figure 9 shows the detailed FD accuracy of individual classes in no noise and various SNR conditions (22 classes: seven types of single and combined bearing faults, three degradation levels for each fault type and normal condition). It can be observed that in a complex scenario, misclassifications can occur in different classes because each fault class depends on various constraint conditions. It is hard to predetermine working conditions to focus on traditional methods such as fault frequency range. Instead, EfficientNet, which can classify up to 1000 classes when it is trained with a large enough dataset like ImageNet, is appropriate to classify each bearing fault class automatically in complex working environments. Figure 10 illustrates the confusion matrix of the fault classification results for Dataset 3.

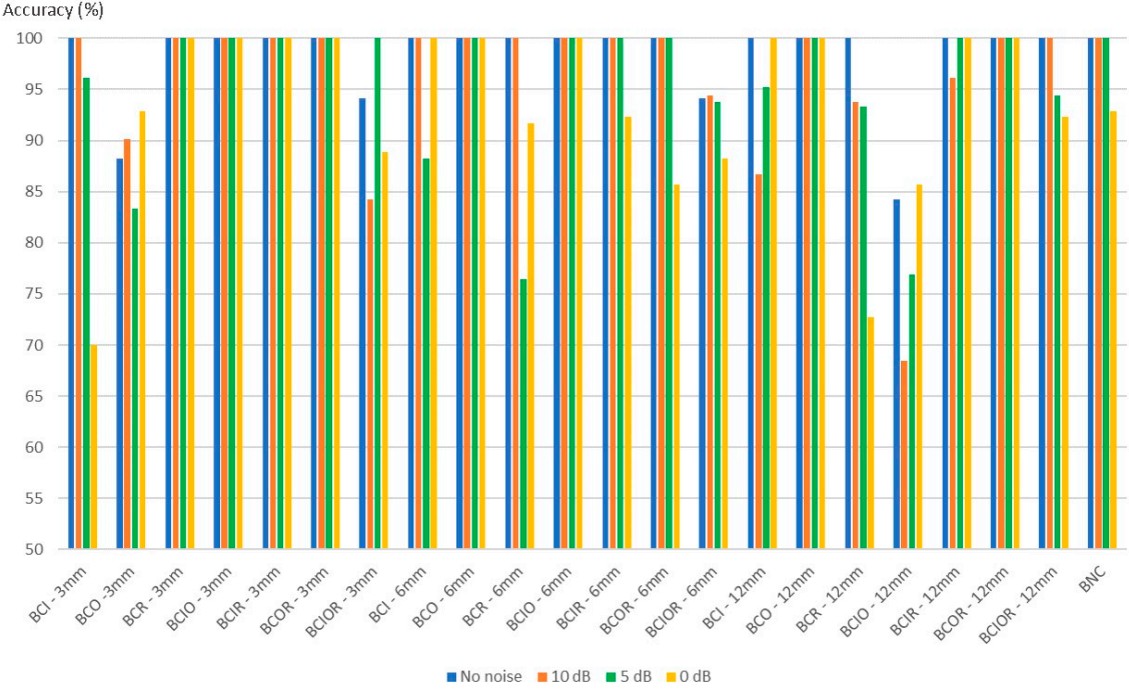

**Figure 9.** Bearing faults diagnosis accuracy for compound faults and different degradation levels.

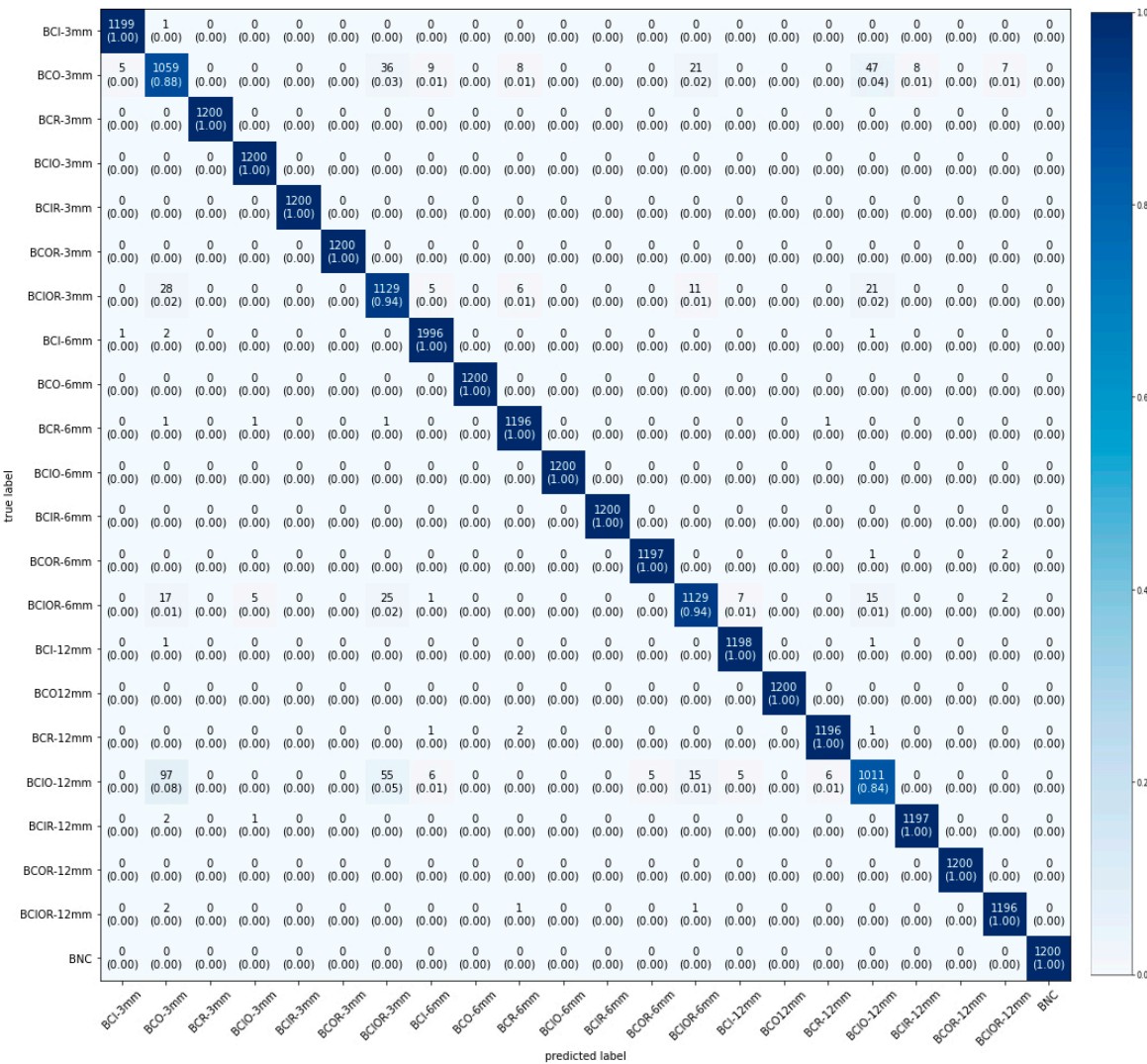

**Figure 10.** Confusion matrix for the testing result of Dataset 3.

## 5. Conclusions

In this paper, we proposed a new CNN-based fault diagnosis method using acoustic emission (AE) signals of rotary-machine bearings. In the proposed method, AE signals acquired from bearings are represented by spectrograms to obtain as much information as possible in the time–frequency domain. Then, feature extraction and classification processes are performed by deep learning using EfficientNet and a stochastic line-search optimizer. According to our experiments for identifying bearing faults under inconsistent and complicated working conditions, the proposed method achieved high accuracy, nearly 100%. We can observe that the transient AE signals' spectrograms can represent fault features well under inconsistent working conditions. Besides, the analysis of time–frequency representations by using state-of-the-art CNN architectures with an SLS optimizer can reduce the number of signal samples and the period required for the training process dramatically (each certain rotational speed, 30 segmented signals in each bearing status are recorded at 3 s duration). Moreover, improving the proposed method's convergence ability proved that it is not only automatic and accurate but also timesaving in signal acquisition and training processes for bearing fault diagnosis. The stability of the proposed method was proven even under ambient noise and in complex scenarios (classifying 22 classes of bearing faults and degradation levels). Therefore, we can expect that the proposed method can be applied widely in real industrial machines for accurate bearing fault diagnosis.

**Author Contributions:** M.T.P.: Software implementation, writing—draft; J.-M.K.: Analysis of results, writing—review and editing; C.H.K.: Idea, validation, analysis of results, and writing—original draft and review. All authors have read and agreed to the published version of the manuscript.

**Funding:** This work was supported by Basic Science Research Program through the National Research Foundation of Korea (NRF) funded by the Ministry of Education (NRF-2018R1A2B6005740).

**Conflicts of Interest:** The authors declare no conflict of interest.

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
