# Peer review of "Intelligent Fault Diagnosis Method Using Acoustic Emission Signals for Bearings under Complex Working Conditions"

_applsci, doi:10.3390/app10207068_

Round 1

Reviewer 1 Report

This paper proposes a method for bearing fault diagnosis based on acoustic emission (AE) signals. The proposed method uses a deep learning method, EfficientNet, to automatically extract fault features from spectrograms of AE signals. The paper explains the proposed method in detail and evaluated through a comprehensive experimental study. Results show the proposed method achieves high diagnosis accuracy under the defined scenarios. The paper is well written with clear structure and detailed explanations.

The author is suggested to consider the following comments.

1) The proposed method works very well under defined scenarios. How to generalize the proposed method for other conditions? Can the trained model be used for other speed conditions, like 800 RPM? If a group of AE signals are sampled with another sampling frequency, can the trained model still work for such AE signals?

2) Page 6, Figure 5(b), the illustrations have covered too much parts of the original image. Suggest to improve this figure.

3) Page 7, it’s difficult to observed the simulated defects in Figure 6. Suggest to enlarge the pictures of the bearings.

4) Suggest the author to give the confusion matrix of the results in Figure 8. This will give a more clear indication of the performance of the proposed method in diagnosing the fault types, especially in classifying the faults with different prediction levels.

Author Response

Please see the attached PDF file containing the summary of changes according to your valuable comments.

Reviewer 2 Report

This paper proposes a methodology for assessing faults in bearings under complex conditions, among which, variable speed and noise are some of them. To be clear, the variable refers to bearings under different speed conditions (and not changes in the speed during operation). In order to simulate this they use a bearing dataset with different speed conditions and combine them to create an evaluation of their method.

Their method brings concepts previously presented in the domain of deep learning to an industrial application. The key points are:

  • AE signals converted into spectrograms.
  • Image augmentation techniques are used to enhance performance of the CNN.
  • EfficientNet Is used as an automatic architecture search.

The results of the paper are hard to assess vs the original benchmark as the rearrangement is original on its own. This part is pending upon the doubts within clarifications are provided.

Overall, the paper has a good structure and good level of English.  The corrections concerning style and grammar are up to the authors to decide if they accept them. With respect of the figures and formatting concerns more the visual experience of the paper. Finally, the clarification section addresses the main points of discussion and must be answered in detail.

Style/Grammar/Missing information

  • I recommend proofreading once more the text and removing some adverbs which do not contribute to a clear description either as they are redundant or emphasize a personal assessment. Here two examples, but there may be others throughout the text, be careful and objective when using these assessments:
  • Line 61. “ AE signals are converted simply into two dimensional spectral energy distribution maps to feed the CNN.

                                           Explanation: Here simply is a personal assessment and does not contribute to making a clear description.

  • Line 68. “ … we propose a new method to achieve very high FD accuracy in diagnosing …”

Explanation: Here very is a personal assessment as this can only be contrasted against other methods. If possible, provide a numerical value from your own results, i.e,  “ achieve high FD (F1-Score 0.95) “

  • Line 83-84. “ the created spectrogram images containing useful information for fault classification are applied to image augmentations “.
    • Explanation: Please rephrase. Image augmentation techniques are applied and not the other way around.
    • Explanation: Perhaps remove containing useful information for classification in order to make the sentence easier to read. We can safely assume that the data is represented as an image for a good reason.
  • Line 94. Change tense. "... that time or frequency domain analysis have to face ..." -> " that time or frequency domain analysis face".
  • Line 110. Use another verb, such as depends on. “Optimizing STFT is related to …”.
  • Line 118. Change “ is set as 1024” -> “is set to 1024“ 
  • Line 118. Add and change. “the hop size is set to wlen as referred to various experiments” ->the hop size is set to wlen/4 as referrer in various experiments”
  • Line 133. The text mentions image argumentation, it should be image augmentation.
  • Line 145. “Scaling up the CNN is broadly implemented to realize superior precision”. This sentence is unclear and may require a citation. So far we have not been introduced to the concept of scaling up an architecture.
    • Change the verb realize as its usage is probably wrong here.
    • Based on https://ai.googleblog.com/2019/05/efficientnet-improving-accuracy-and.html, the aim of scaling up is to start from a base architecture and try more complex ones.
  • Line 147. A word is missing “...existing CNN structure is defined by function” -> “...existing CNN structure is defined by the function”
  • Line 173. The first sentence is redundant, rephrase it to make it clear. In the CNN model's training stage, Gradient Descent (GD) is one of the most well-known and 174 standard algorithms to optimize the CNN.
  • Line 235. This sentence is unclear. What are the hidden characteristics? “Test subset 234 and validation subset are the same hidden characteristic and not used to update model parameters 235 while training.”

Figures

  • Figure 1.
    • Use a higher resolution image.
    • After the second step text says "image argumentation", it should be image
  • Figure 2.
    • The legend for the color map is unreadable.
    • The axes of the images are unreadable.
    • I think the subcaption (a) (b) (c) seems to be aligned incorrectly.
  • Figure 3.
    • Use a higher resolution image
  • Figure 4.
    • This figure is a copy (screenshot) from the original source of EficientNet. Please rewrite it so it has a better resolution.
  • Figure 5.
    • Use a higher resolution image.
  • Equation 3.
    • Use a higher resolution image.

Formatting

  • Line 51. Please adjust or change the way how time-frequency is abbreviated as it disaligns the vertical spacing.
  • Line 87. The abbreviation SLS was previously introduced in line 73, hence there is no need of introducing it here again.
  • Line 118. There seems to be an extra white space before the symbol wlen
  • Line 147-148. The equations in the text seem to cause an extra vertical spacing. Verify if this can be fixed.

Clarification

  • Line 65. “With the proven efficiency of CNN-based methods … “
    • Explanation: This sentence is unclear. If CNN-based methods have proven to be efficient, then what is the contribution? Which are the characteristics of a more suitable CNN model? Are you overcoming some of the problems that previous CNN-based architectures have?
  • Concerning the image augmentation (IMPORTANT)
    • As this seems to be a key step in your method, it is important to justify why it is used. For example, during your introduction you should refer to CNN based methods which have used this technique previously for classification tasks.
    • Please include a description of the augmentation techniques that were used.
    • Image augmentations make sense in visual images where a transformation would produce still a viable image. For example, a translation in an image is valid as the objects are still present in the image. However, a translation in an spectrogram would cause the amplitude in a given time or frequency to shift, would this still be a valid source of information?
    • Line 134. Using image augmentation creates more images, which indeed can reduce the number of samples needed for training. For example, according to https://journalofbigdata.springeropen.com/articles/10.1186/s40537-019-0197-0, image augmentation increases training time and memory usage.
  • EfficientNet
    • Line 147. The equation contains two times L_{i}, the first one as a super index to indicate the number of repetitions and a second time to indicate the second dimension of the tensor. The equation is wrong and differs from the source https://arxiv.org/pdf/1905.11946.pdf.
  • Choice of gradient method
    • Line 178. The text addresses that different optimizers need to be tested before selecting one and concludes that Adamax is the optimal. Is this something you evaluated or that the source [20] did? In case that source [20] was, it is important that you worked on the same dataset.
    • Line 180. This sentence is unclear. “ Stochastic gradient descent (SGD) and its variants take precedence optimization methods in 181 modern machine learning”
    • Line 183. This segment mentions the full slope calculation methods, but these haven’t been explained.
  • Experimental results
    • Line 253. Please specify the exact number of repetitions, and if you consider it accordingly report the standard deviation or confidence interval. “ All of the experiments on datasets are repeated for about ten times.”
    • (IMPORTANT)
      • From the provided description is unclear what is the failure rate (as baseline).
      • If I understand this correctly, the original dataset was regrouped in sets that contained different rotational speeds (as mentioned in Table 3). If this is the case, this needs to be clarified explicitly at the beginning of section 3. If not, a reader will look into source [3] and see there is a difference between the 3 mentioned datasets here and the 10 of the original material (see https://ieeexplore.ieee.org/stamp/stamp.jsp? tp=&arnumber=7403935&tag=1). (As in comparing Dataset 1 from your publication vs Dataset 1 of [3]).
      • The original source does not report a 250 RPM dataset. Could you please clarify?
    • Was the validation data used for Early Stopping or what type of validation was used?
    • Table 2.
      • Which was the ratio and the procedure for splitting the validation data?
    • Line 285. Mentioning the number of epochs does not give a clear idea of the fast convergence w.r.t. time. Given the additional step of image augmentation (as previously mentioned) it is important to compare the clock-time. In addition, could you specify in the image augmentation stage, how many additional images are passed per epoch?
    • Line 293. The higher accuracy may not be attributed by removing the need of selecting a correct learning rate. Not having to select remains an advantage, but is hard to prove is the reason the model does not converge to the same accuracy level.
    • EfficientNet is missing. Although an automatic architecture search was used, the results on the architecture search were not presented.

Author Response

(The authors gave the same response as above.)

Round 2

Reviewer 2 Report

Thanks for addressing the doubts on the paper, most importantly the remarks concerning the data normalization.

The results look correct and the publication seems to be innovative.

A last group of remarks concerning the text correction:

  1. face -> phase
  2. characteristics which is unseen by the model and is not used to update 229 model parameters while training -> characteristics which are unseen by the model and are not used to update the model parameters while training
  3. are repeated for ten times